Spatial distribution of microbial communities among colonies and genotypes in nursery-reared Acropora cervicornis

Miller Nicole 1
Maneval Paul 1 2
Manfrino Carrie 2
Frazer Thomas K. 1
Meyer Julie L. juliemeyer@ufl.edu 3
1 School of Natural Resources and Environment, University of Florida , Gainesville , FL , United States of America
2 Little Cayman Research Center, Central Caribbean Marine Institute , Little Cayman , Cayman Islands
3 Soil and Water Sciences Department, Genetics Institute, University of Florida , Gainesville , FL , United States of America
Rappe Michael
Electronic publication date: 2020 Aug 26
Publication date: 2020
Volume: 8
Electronic Location ID: e9635
Received 2020 May 5; Accepted 2020 Jul 9
Copyright: ©2020 Miller et al.
Copyright year: 2020
Copyright holder: Miller et al.
License: This is an open access article distributed under the terms of the Creative Commons Attribution License, which permits unrestricted use, distribution, reproduction and adaptation in any medium and for any purpose provided that it is properly attributed. For attribution, the original author(s), title, publication source (PeerJ) and either DOI or URL of the article must be cited.
License URL: https://creativecommons.org/licenses/by/4.0/

Keywords: Acropora cervicornis, Microbiome, Coral microbiology, Coral restoration

Funding: USDA-NIFA Hatch FLA-SWS-005474 Disney Conservation Fund Dart Foundation Ernest Kleinwort Charitable Trust Consolidated Water Company Sample collection and molecular lab work was supported by a USDA-NIFA Hatch fund project FLA-SWS-005474 to Julie Meyer. Funding for field work was provided to the Central Caribbean Marine Institute’s Coral Reef Resilience and Restoration initiative (http://reefresearch.org) by the Disney Conservation Fund, the Dart Foundation, the Ernest Kleinwort Charitable Trust, and the Consolidated Water Company. The funders had no role in study design, data collection and analysis, decision to publish, or preparation of the manuscript.

==============================
Background

The architecturally important coral species Acropora cervicornis and A. palmata were historically common in the Caribbean, but have declined precipitously since the early 1980s. Substantial resources are currently being dedicated to coral gardening and the subsequent outplanting of asexually reproduced colonies of Acropora, activities that provide abundant biomass for both restoration efforts and for experimental studies to better understand the ecology of these critically endangered coral species.

Methods

We characterized the bacterial and archaeal community composition of A. cervicornis corals in a Caribbean nursery to determine the heterogeneity of the microbiome within and among colonies. Samples were taken from three distinct locations (basal branch, intermediate branch, and branch tip) from colonies of three different coral genotypes.

Results

Overall, microbial community composition was similar among colonies due to high relative abundances of the Rickettsiales genus MD3-55 (Candidatus Aquarickettsia) in nearly all samples. While microbial communities were not different among locations within the same colony, they were significantly different between coral genotypes. These findings suggest that sampling from any one location on a coral host is likely to provide a representative sample of the microbial community for the entire colony. Our results also suggest that subtle differences in microbiome composition may be influenced by the coral host, where different coral genotypes host slightly different microbiomes. Finally, this study provides baseline data for future studies seeking to understand the microbiome of nursery-reared A. cervicornis and its roles in coral health, adaptability, and resilience.

Introduction

Historically, Acropora cervicornis and A. palmata corals were the most abundant corals throughout the broader Caribbean region (Goreau, 1959; Bellwood et al., 2004; Pandolfi & Jackson, 2006). Since 1980, an estimated 97% of native populations of Acropora corals have diminished throughout the Caribbean (Boulon et al., 2005), with large losses attributed to the microbially mediated white-band disease (Aronson & Precht, 2001). Both A. cervicornis and A. palmata are now listed as critically endangered by the IUCN (Aronson et al., 2008) and threatened under the Endangered Species Act (Hogarth, 2006), providing a legal mandate to facilitate the recovery of these keystone species. Ecologically, repopulation of Acropora corals is critical for improving the health of entire reef systems because they contribute to reef accretion and provide structural complexity that serves as important habitat for fish and other reef organisms.

In response to the loss of coral reefs, organizations around the Caribbean have turned to coral gardening to replenish local populations of coral (Bayraktarov et al., 2020; Young, Schopmeyer & Lirman, 2012). For example, coral restoration projects have resulted in the outplanting of tens of thousands of nursery-reared Acropora corals on Florida reefs each year (Schopmeyer et al., 2017; Lirman & Schopmeyer, 2016). These restoration efforts rely primarily on the asexual reproduction of these corals as small coral fragments grow at faster rates than larger colonies (Lirman et al., 2014) and sequential fragmentation, which mimics natural breakage cycles, can provide more new coral colonies for restoration than could be produced through sexual reproduction in the same time frame. Fragmented corals rapidly grow in in situ ocean nurseries and are then outplanted back to reefs, a method that has proven to be highly effective for producing abundant A. cervicornis biomass. This improved growth and decreased mortality of A. cervicornis in ocean nurseries in comparison corals attached to reefs is due in part to reduced predation by the corallivorous snail Coralliophila abbreviata and the fireworm Hermodice carunculata (Young, Schopmeyer & Lirman, 2012). In addition to the direct effects of predation on colony health, C. abbreviata is a known vector for white-band disease (Gignoux-Wolfsohn, Marks & Vollmer, 2012) and H. carunculata is a known vector for the coral pathogen Vibrio shilohi (Sussman et al., 2003).

However, the survival of Acropora corals once placed back on the reef is often lower than desired given the resource-intensive nature of nursery production (Mercado-Molina, Ruiz-Diaz & Sabat, 2014; Ladd et al., 2016; Lohr & Patterson, 2017). Survival of restored Acropora is also highly variable among sites (Schopmeyer et al., 2017; Johnson, 2015), suggesting that some environments may be better than others for long-term restoration efforts. To date, research has focused on optimizing out-planting techniques and measuring performance of different coral genotypes (Lirman et al., 2014; Miller et al., 2019; Ladd et al., 2017; Goergen & Gilliam, 2018; Goergen, Ostroff & Gilliam, 2018), while few studies have examined the microbial community composition associated with the success of outplanted corals. Given the critical role of the microbiome in both the health and disease of corals (Bourne et al., 2009; Bourne, Morrow & Webster, 2016; Krediet et al., 2013), manipulation of the coral microbiome has recently been proposed as a way to increase the resilience of corals (Peixoto et al., 2017; Rosado et al., 2018; Rosenberg et al., 2007) and improve restoration efforts (Van Oppen et al., 2017; West et al., 2019). Several recent review papers (Bourne, Morrow & Webster, 2016; Peixoto et al., 2017; Van Oppen & Blackall, 2019; Sweet & Bulling, 2017; Rädecker et al., 2015) have detailed the potential roles that the microbiome plays in the overall health of the coral holobiont, including protection against pathogens, tight recycling of nutrients within the holobiont, and nitrogen fixation, which benefits the photosynthetic dinoflagellate symbionts, and there is growing interest in designer microbes and probiotic strains to mitigate loss of coral reefs (Peixoto et al., 2017; Rosado et al., 2018; Peixoto, Sweet & Bourne, 2019; Damjanovic et al., 2017).

Even as our ability to characterize coral microbial communities has become exponentially faster and cheaper due to technological advances in DNA sequencing, the availability of natural corals for study has drastically decreased. Leveraging access to abundant biomass in nursery-reared corals with known genotypes, we address two basic questions about the spatial distribution of coral microbial communities. First, we assess whether a single sample is representative of the microbiota across an entire colony, information that is useful for both researchers and resource managers that strive to reduce stressors to already threatened species. To date, only a handful of studies have assessed spatial distribution of the microbial communities within coral colonies. Spatial heterogeneity was observed using bacterial community fingerprinting methods in branching Porites furcata between branch tips and branch bases (Rohwer et al., 2002) and across mounding Orbicella annularis corals (Daniels et al., 2011). In contrast, pyrosequencing in A. palmata showed no statistical difference in bacterial communities from the top, underside, and base of the colonies, regions with distinct irradiance levels (Kemp et al., 2015). Second, we address whether the microbial community changes with coral genotype. Given the limited number of extant natural populations of A. cervicornis in many parts of the Caribbean, there are limited studies of its microbial community composition. Previous studies of natural populations have shown that the microbiota of A. cervicornis in Puerto Rico varied between deep and shallow sites (Godoy-Vitorino et al., 2017) and that intercolony variation of A. cervicornis in Panama was weaker than seasonal variation (Chu & Vollmer, 2016). However, variation of bacterial communities in A. cervicornis from Panama used in a disease transmission study demonstrated that colony had a stronger effect than collection site (Gignoux-Wolfsohn, Aronson & Vollmer, 2017). Here, we apply an extensive sampling scheme to A. cervicornis in ocean nurseries in the Cayman Islands to examine the spatial distribution and heterogeneity of A. cervicornis microbial communities.

Materials & Methods

Sample collection

Field collections were authorized by permit from the Department of Environment on behalf of the National Conservation Council, Cayman Islands to CM and TF (December 1, 2017 to December 1, 2018). Coral colonies for this study were sampled at the Integrated Coral Observation Network Reef Nursery (−80°3′36″N, 19°41′60″E) operated by the Central Caribbean Marine Institute (CCMI) in Little Cayman. The nursery uses a coral gardening technique (Lirman & Schopmeyer, 2016; Lirman et al., 2014) on PVC trees and frames to cultivate A. cervicornis at 18 m depth, under conditions of natural currents, temperatures, and light. Three previously identified coral genotypes determined via genotyping-by-sequencing (Drury et al., 2017), herein referred to as green (G), red (R), and yellow (Y) genotypes, were sampled from the nursery. Colonies of these genotypes were added to the CCMI nursery in 2012 as fragments from local populations. Each colony was fragmented from a larger coral colony in 2016, approximately one and a half years prior to sampling. Colonies used in this study had no apparent signs of disease or illness at the time of sample collection. Colonies observed 10 months after sampling showed no signs of disease or distress (Fig. 1).

Figure 1 Nursery-reared Acropora cervicornis sampled in December 2017 for microbial community composition.

(A) Three colonies each of red (R), green (G), and yellow (Y) coral genotypes were sampled. (B) On each colony, three replicate samples were collected on basal branches (1°), intermediate branches (2°), and apical colony tips (3°). (C) The same colonies photographed in October 2018 showed rapid growth and no signs of disease. Acropora cervicornis icon courtesy of the Integration and Application Network, University of Maryland Center for Environmental Science (ian.umces.edu/symbols/).

All in situ collection was completed on December 8, 2017 by AAUS-certified science divers. The coral microbial community was sampled from three colonies each of three coral genotypes. All 9 colonies hung within 1 meter of each other on the same frame structure (Fig. 1). On each colony, three replicate samples were collected from each of the following locations: basal branches (1°), intermediate branches (2°), and apical colony tips (3°), for a total of 81 coral surface samples. After brief agitation by a needleless syringe, the surface mucus layer and coral tissue was aspirated with a sterile 60-ml syringe. Collections were transported back to the CCMI laboratory within the hour. Syringes were positioned vertically for about five minutes to allow the coral sample material to settle at the tip of the syringe. Coral mucus and tissues were transferred to sterile 2-ml cryotubes, centrifuged at 13,000 rpm for 5 min, and the saltwater supernatant was decanted. Lastly, 5 volumes of RNAlater (Ambion, Austin, TX) to 1 volume sample pellet was added to the cryotube. Samples were incubated at 4 °C for 12 h and then stored at −20 °C until nucleic acid extraction.

V4 amplicon library preparation

DNA was extracted from up to 0.5 ml of mucus and tissue using a DNeasy Powersoil Kit (Qiagen, Germantown, MD) according to the manufacturer’s instructions, including bead beating for 10 min. As extraction controls were not collected when samples were processed in early 2018, we acknowledge the potential for contamination from lab reagents (Salter et al., 2014), either in the extraction kit or in subsequent PCR and cleanup reagents. The processing of extraction controls has since become standard practice in our lab beginning in mid-2018. After DNA extraction, guidelines of the Earth Microbiome Project 16S Illumina Amplicon Sequencing Protocol (Caporaso et al., 2011) were followed for amplicon sequencing preparation using the 2015-current primer set with barcoded forward primer 515FB (Parada, Needham & Fuhrman, 2016) and reverse primer 806RB (Apprill et al., 2015), targeting Bacteria and Archaea. The V4 region of the 16S rRNA gene was amplified in three replicate 25-µl reactions for each coral sample using Phusion High-Fidelity PCR Master Mix (New England Biolabs, Ipswich, MA), 3% dimethyl sulfide, 0.25 µM of each primer, and roughly 10 ng of DNA template. PCR cycling was as follows (35 cycles): 45 s at 94 °C denaturation, 60 s at 50 °C annealing, 90 s at 72 °C extension. Initial denaturation and final extension were for 3 min at 94 °C and 10 min at 72 °C, respectively. Negative PCR controls were examined by gel electrophoresis on ethidium bromide-stained 1% agarose gels to ensure no contamination. Triplicate PCR products were then combined and purified using a MinElute PCR Purification Kit (Qiagen). Cleaned amplicon library concentrations were determined with a Nanodrop 1000. A total of 240 ng of each amplicon library was pooled for sequencing using an Illumina MiSeq with paired 150-bp reads (v.2 cycle format) at the University of Florida Interdisciplinary Center for Biotechnology Research.

Analysis of V4 amplicon libraries

Illumina adapters and primers were removed using cutadapt v. 1.8.1 (Martin, 2011). Additional processing and analysis of amplicon libraries was completed in R v. 3.5.1. Quality filtering, error estimation, merging of reads, dereplication, removal of chimeras, and selection of amplicon sequence variants (ASVs) were performed with DADA2 v. 1.10.0 (Callahan et al., 2016) as in (Meyer et al., 2019), using the filtering parameters truncLen=c(150,150), maxN=0, maxEE=c(2,2), truncQ=2, rm.phix=TRUE to remove all sequences with ambiguous basecalls and phiX contamination. Taxonomy was assigned in DADA2 to ASVs using the SILVA reference dataset v. 132 (Yilmaz et al., 2014). Sequences that could not be assigned as bacteria or archaea and sequences identified as chloroplasts or mitochondria were removed from further analysis.

The ASV and taxonomy tables, along with associated sample metadata were imported into phyloseq v. 1.26.0 (McMurdie & Holmes, 2013) for community analysis. ASVs with counts across all samples of less than five were removed from the analysis. ASVs with zero counts were transformed using the count zero multiplicative method with the zCompositions package v. 1.1.2 in R (Palarea-Albaladejo & Martín-Fernández, 2015). The zero-replaced read counts were transformed with the centered log-ratio transformation and the Aitchison distance metric was calculated with CoDaSeq v. 0.99.2 (Gloor et al., 2016). Principal component analysis of the Aitchison distance was performed with the package prcomp in R and plotted with ggplot2 v. 3.1.0 (Wickham, 2016). Analysis of similarities (ANOSIM) was performed on the Aitchison distance using the R package vegan v. 2.5-3 (Dixon, 2003). Differential abundance of microbial families among coral genotypes was determined with Analysis of Composition of Microbiomes (ANCOM) (Mandal et al., 2015) as in (Meyer et al., 2019), using an ANOVA significance level of 0.05 and removing families with zero counts in 90% or more of samples. Only families detected in at least 70% of samples were reported. All metadata and R scripts used for analysis are available in GitHub (https://github.com/meyermicrobiolab/Microbiomes-of-nursery-reared-Acropora-cervicornis).

Results

A total of 81 amplicon libraries were analyzed from nine separate coral colonies representing three genotypes of A. cervicornis. After quality control, libraries contained an average of 77,923 reads and ranged from 432-184,540 reads (Table S1). Sequencing reads with primers and adapters removed are available at NCBI’s Sequence Read Archive under BioProject PRJNA495377. Overall, nursery-reared A. cervicornis microbial communities were similar within colonies and did not differ statistically by branch location (ANOSIM R = 0.024, p = 0.057). Microbial community composition varied by both coral genotype (ANOSIM R = 0.199, p = 0.001) and by colony (ANOSIM R = 0.264, p = 0.001), although the effect size of each was relatively small. Microbial communities of yellow colonies appeared more distinct from microbial communities of the red and green coral genotypes (Fig. 2).

Figure 2 Principal Component Analysis of the Aitchison distance between microbial communities from green (G), red (R), and yellow (Y) genotypes of nursery-reared Acropora cervicornis.

Location of samples within individual colonies are indicated by symbol shape. Principal Component 1 (PC1) explained 24% of the variation among communities and Principal Component 2 (PC2) explained 8% of the variation among communities.

The most abundant bacterial orders detected were Rickettsiales, Synechococcales, Vibrionales, and an unclassified order of Alphaproteobacteria (Fig. 3). Both Rickettsiales and Synechococcales were detected in most samples. Diverse ASVs classified as Vibrionales were common and most abundant in the yellow genotype colonies. One abundant ASV from the unclassified order of Alphaproteobacteria common in corals of the green genotype was an exact match to an unpublished clone library sequence from A. palmata in Puerto Rico (GenBank Accession EU853842) and 99.6% similar to a clone sequence (GenBank Accession AY323179) from healthy A. palmata in Barbados (Pantos & Bythell, 2006).

Figure 3 Relative abundance of amplicon sequence variants of the V4 region of 16S rRNA genes in nursery-reared Acropora cervicornis, colored by bacterial order.

Samples are grouped by coral genotype: (A) green (G), (B) red (R), and (C) yellow (Y) and by location within the colony: basal branches (1°), intermediate branches (2°), and apical colony tips (3°).

There were two samples with very distinct community structure: one sample from a basal branch (1°) on a red genotype colony and one sample from an apical tip (3°) on a yellow genotype colony (Figs. 3A, 3C). The unique sample taken from a red colony had a relative abundance of 38% for a single Alteromonadales ASV classified as Algicola, which was an exact match to a clone library sequence from a diseased Pavona duerdeni coral in Thailand (GenBank Accession KC527315) (Roder et al., 2014) as well as to several other marine sources. The unique sample taken from a yellow colony also had high relative abundances of two Alteromonadales ASVs, which were different from the predominant Alteromonadales ASV in the red colony. The first was an ASV classified as Thalassotalea (16% relative abundance) that was an exact match to the type strain Thalassotalea euphylliae str. Eup-16 isolated from the coral Euphyllia glabrescens (GenBank Accession NR_153727.1) (Sheu et al., 2016) and to clone libraries from the coral Galaxea fascicularis (GenBank Accession KU354186), crustose coralline algae (GenBank Accession JQ178640.1) (Webster et al., 2013), and the sponge Diacarnus erythraeanus (GenBank Accession HM854473.1) (Bergman et al., 2011). The second common Alteromonadales ASV in the red colony was an unclassified genus of Alteromonadaceae (11% relative abundance) that was an exact match to several clone library sequences from corals, including Gorgonia ventalina (GenBank Accession GU118325) (Sunagawa, Woodley & Medina, 2010), Porites lutea (GenBank Accession KF179648) (Séré et al., 2013), Acropora pruinosa (GenBank Accession JQ347406), and a coral-encrusting sponge (GenBank Accession HM593584) (Tang et al., 2011). The unique yellow colony sample was also distinguished by the high relative abundance (27%) of one Opitutales ASV classified as Coraliomargarita and was 98% similar to clone library sequences from the corals Porites lutea (GenBank Accession KF179779) (Séré et al., 2013) and Favia corals with Black Band Disease (GenBank Accession GU472376).

By far, the most abundant bacterial order in nearly all of the samples from this study was Rickettsiales. In total, there were 11 ASVs classified as Rickettsiales, including five ASVs classified as Rickettsiales genus MD3-55, one ASV classified as Rickettsiales family S25-593, and five ASVs classified as Rickettsiaceae. The two most abundant ASVs were both classified as genus MD3-55 and together these two ASVs averaged a relative abundance of 75% in all samples. The remaining 9 Rickettsiales ASVs collectively had an average relative abundance well below 1%. The two most abundant MD3-55 ASVs (hereafter, ASV1 and ASV2, respectively) differed by a single nucleotide. ASV1 differed by one nucleotide in the V4 region of the 16S ribosomal RNA gene from the metagenome-assembled genome (GenBank Accession NZ_RXFM01000000) of an MD3-55 population from A. cervicornis collected in the Florida keys named Candidatus Aquarickettsia rohweri strain a_cerv_44 (Klinges et al., 2019). ASV2 was an exact match in the V4 region to the metagenome-assembled genome. According to BLASTn results, both abundant Rickettsiales ASVs in all samples were also 99%–100% identical to a clone library sequence from healthy Caribbean Orbicella faveolata (GenBank Accession JQ516457) (Kimes et al., 2013).

The distribution of the five Rickettsiales genus MD3-55 ASVs was strikingly different across the three coral genotypes (Fig. 4). Only ASV1 was detected in all samples from the three green genotype colonies and all samples from one red genotype colony. The other two red colonies were predominately ASV2, along with lower levels of ASV3 and ASV4. All five of the MD3-55 ASVs were detected in the yellow colonies, which contained predominantly ASV2. To determine the consistency of this result, we repeated the entire analysis pipeline six times to calculate the relative abundance of the five MD3-55 ASVs across all samples and achieved identical results.

Figure 4 Proportion of amplicon sequence variants classified as the Rickettsiales genus MD3-55 relative to all ASVs in the communities in colonies of green (G), red (R), and yellow (Y) coral genotypes of nursery-reared Acropora cervicornis.

Each bar represents one sample, with a total of nine samples per coral colony.

Given the high relative abundance of just two Rickettsiales ASVs across nearly all samples, the differences in microbial community composition among coral genotypes were based on microbes with low relative abundances. Sixteen microbial families were differentially abundant across coral genotypes (Fig. 5). The largest difference among coral genotypes was in an unclassified family of Alphaproteobacteria (6 ASVs), corresponding to the unclassified order of Alphaproteobacteria that was common in green colonies (Fig. 3). This Alphaproteobacteria family was most abundant in green colonies and least abundant in red colonies (Fig. 5). As described above, the most abundant ASV in this family matched sequences previously detected in coral microbial communities. In contrast, the family Francisellaceae (4 ASVs) was most abundant in red colonies and least abundant in green colonies. The most abundant Francisellaceae ASV was 99.6% similar to four clone library sequences (GenBank Accessions FJ202895, FJ202734, FJ202359, FJ202230) and more than 98% similar to twenty additional clone library sequences from the same study on Caribbean Orbicella faveolata corals (Sunagawa et al., 2009).

Figure 5 Relative abundance of amplicon sequence variants in differentially abundant families between colonies of green (G), red (R), and yellow (Y) coral genotypes of nursery-reared Acropora cervicornis.

To increase separation of very small values, relative abundances are plotted on a log scale, with 0.001 added to every value to avoid taking the log of zero.

The microbial community composition of yellow colonies appeared distinct from both red and green colonies (Fig. 2), which appears to be driven primarily by four differentially abundant bacterial families that were detected in higher abundances in the yellow colonies compared to red and green. These include Desulfobacteraceae, Pirellulaceae, Puniceicoccaceae, and Xenococcaceae (Fig. 5). A total of four Desulfobacteraceae ASVs were detected in this study. The most abundant Desulfobacteraceae ASV was an exact match to clone library sequences from several coastal marsh habitats, including a clone library sequence (KF513059) associated with the “pink berry” consortia of sulfate-reducing bacteria and sulfur-oxidizing bacteria (Wilbanks et al., 2014). A total of two Pirellulaceae ASVs (phylum Planctomycetes) were detected at low abundance in most samples and these sequences matched clone library sequences from a variety of marine habitats. There were eight Puniceicoccaceae ASVs (phylum Verrucomicrobia) and the most abundant were classified as Coraliomargarita as discussed above and Lentimonas. The most abundant Lentimonas ASV was an exact match to many clone library sequences from marine habitats, including an unpublished sequence (MF039937) from the sponge Theonella. A total of four Xenococcaceae ASVs (phylum Cyanobacteria) were detected and the most abundant was almost entirely absent from red and green colonies. This ASV was classified as the genus Xenococcus PCC-7305 and was an exact match to clone library sequences from stromatolites in the Bahamas (EU249096) (Foster et al., 2009) and several unpublished clone library sequences from ooilitic sands in the Bahamas (JX255853, JX255856, JX255863, JX255892, JX504367, JX504398).

Discussion

By characterizing the microbial community composition of nine samples per coral colony, we have demonstrated that sampling from one location on an A. cervicornis coral colony is likely to provide a representative sample of the microbial community for the entire colony. This has important implications for researchers and resource managers who are concerned with how many samples are appropriate to take per colony and how to minimize sampling to reduce stress to colonies. It is also important to note that even with the heavy sampling that was performed here, namely nine samples taken in one day from a basketball-sized colony, no visible stress to the colonies was discernable. These colonies were observed during regular maintenance of the in situ nursery and roughly a year after sampling, all of the colonies were thriving. Our finding that one sample is likely representative of the whole colony is consistent with a similar analysis of A. palmata colonies that demonstrated microbial communities were not different between the topside of the wide branches of elkhorn coral and the more shaded underside of branches and colony bases (Kemp et al., 2015). Thus, for Caribbean acroporid corals, it appears that microbial community composition is relatively uniform at the colony scale. Since both of these coral species are critically endangered, this means that sampling for microbiota can be minimized to reduce impact to the coral host. However, this remains to be tested more broadly across different coral species, and especially in wild populations when available, rather than the relatively sheltered nursery-reared colonies sampled here.

These results seemingly conflict with earlier work using fingerprinting methods that detected spatial heterogeneity within colonies of Porites and Orbicella corals (Rohwer et al., 2002; Daniels et al., 2011). It is possible that other coral groups exhibit more heterogeneity across colonies, however, this conflict may also be an artefact of the different methods used and the increased sample sizes of our study. For example, Rohwer et al. (2002) used T-RFLP on a total of eight samples and Daniels et al. (2011) used ARISA on a total of seventeen samples. While we did detect differences among samples (and a few samples were clearly very distinct), overall the differences among colonies and among genotypes were stronger than the differences within colonies. Therefore, heterogeneity does exist in the microbial communities across the surface of the coral colony regardless of coral species, but these differences are likely much smaller than differences among colonies.

Strikingly, the most abundantly detected sequences in this study belong to the Rickettsiales genus MD3-55. This is consistent with earlier work on natural populations of A. cervicornis, A. palmata, and A. prolifera at sites in Puerto Rico, Panama, and the Florida Keys (Casas et al., 2004). Casas et al.(2004) found high relative abundances of coral-associated Rickettsiales in clone libraries of near full-length 16S rRNA genes in both A. cervicornis and A. prolifera but were only able to detect Rickettsiales using a targeted nested-PCR approach in A. palmata. Rickettsiales were also detected at low levels in natural populations of A. cervicornis and A. palmata in Panama using metabarcoding of the V6 hypervariable region (Chu & Vollmer, 2016). More recent studies using metabarcoding of the V4 hypervariable region have detected very high relative abundances of Rickettsiales genus MD3-55 in natural populations of A. cervicornis in Puerto Rico (Godoy-Vitorino et al., 2017) and in nursery-reared A. cervicornis (but not in A. palmata) from the Florida Keys (Rosales et al., 2019). In Florida nursery-reared A. cervicornis, experimental nutrient enrichment substantially increased the relative abundance of one Operational Taxonomic Unit of Rickettsiales (Shaver et al., 2017). Together, these results suggest that the Rickettsiales genus MD3-55 is common in Caribbean A. cervicornis and less common in A. palmata. The results also suggest that the predominance of MD3-55 may be impacted by the choice of PCR primer (V6 versus V4, for example) or that variation in the abundance of the presumably intracellular MD3-55 may be influenced by environmental conditions.

The role of the Rickettsiales genus MD3-55 in the health of acroporid corals is unclear. Here, we detected high relative abundances of this genus in all samples from nursery-reared colonies that exhibited no signs of disease in the year and a half prior to sampling and more than a year after sampling. However, this group was originally suspected to play a role in coral disease when intracellular bacterial aggregates were detected in histological slides of Acropora tissues with white-band disease (WBD) (Peters, Oprandy & Yevich, 1983). Subsequently, Rickettisales-like organisms, which can be detected by Geimsa stain, were detectable inside mucocytes of both healthy A. cervicornis and in WBD tissues (Miller et al., 2014; Gignoux-Wolfsohn et al., 2020). The presence of Rickettsiales in association with WBD in A. cervicornis has been confirmed by molecular methods, however, the relative abundance of Rickettsiales at times increased with the disease (Gignoux-Wolfsohn & Vollmer, 2015), and at times did not increase with the disease (Glasl et al., 2019). The recent recovery of a metagenome-assembled genome of the dominant MD3-55 strain detected by Casas et al. (2004) and the widespread detection of MD3-55 sequences in both environmental samples and coral-associated microbial community datasets (Klinges et al., 2019) have shed new light on the role of this apparently ubiquitous intracellular invertebrate symbiont.

The representative genome of MD3-55 from Caribbean A. cervicornis, dubbed Candidatus Aquarickettsia rohweri, has a reduced genome size, limited capacity for the biosynthesis of sugars and amino acids, and genes encoding nucleotide transport proteins, all of which are reflective of its intracellular, host-dependent lifestyle (Klinges et al., 2019). The Candidatus Aquarickettsia rohweri genome also has genes to sense extracellular nitrate levels, despite lacking genes to metabolize nitrogen. It has been hypothesized that Candidatus Aquarickettsia rohweri responds to environmental nitrate enrichment with increased growth, which saps the host of energy and makes it more susceptible to diseases like WBD (Klinges et al., 2019). However, to date strong empirical evidence is lacking for the connection between abundance of members of the genus MD3-55 (Candidatus Aquarickettsia) and quantifiable disease signs. This may, in part, be due to biases in the production of 16S rRNA gene amplicon libraries which may not correspond to true biological abundances in situ or it may reflect the indirect relationship between nutrient enrichment, abundance of MD3-55, and signs of disease in which a certain threshold of MD3-55 must be reached before host health is negatively affected.

Alternatively, there is the possibility that not all strains of MD3-55 are equally harmful to the host. Here, we detected two predominant strains of MD3-55 that varied across coral genotypes in addition to three minor strains. This is consistent with recent work demonstrating that while the genus Endozoicomonas was predominant among Acropora tenuis bacterial communities, individual ASVs of Endozoicomonas were highly host genotype-specific (Gignoux-Wolfsohn & Vollmer, 2015). Similarly, Endozoicomonas strains were identified as sporadic symbionts in the Pacific Line Islands (Hester et al., 2016), where Endozoicomonas was found on Porites lutea from only one island out of six islands sampled. However, further study is required to determine if these strains have genomic differences between strains that result in functional differences that affect coral health.

Conclusions

We examined the heterogeneity of microbial communities within and among coral colonies and found that while some heterogeneity exists within samples from the same coral colony, only differences among coral colonies and genotypes were statistically significant. However, the effect size was small, as almost every sample had high relative abundances of sequences classified as the Rickettsiales genus MD3-55 (Candidatus Aquarickettsia rohweri). To date, this presumably parasitic intracellular bacterium has been associated with white-band disease, although direct evidence of a negative relationship between genus MD3-55 and its coral hosts is lacking. Here, high relative abundances of MD3-55 were found in corals showing no signs of diseases over more than 2 years of monitoring. In addition, we detected a striking pattern of unique amplicon sequence variants of MD3-55 associated with different coral genotypes, suggesting that these strains may have functional differences selected by the host. Future work incorporating both physiological and genomic analysis of these bacteria is required to fully elucidate the role of MD3-55 bacteria in A. cervicornis corals.

Supplemental Information

Supplemental Information 1 List of samples with colony metadata, number of sequencing reads, and GenBank accession numbers

Click here for additional data file.

Additional Information and Declarations

Competing Interests

Author Contributions

DNA Deposition

Data Availability

The authors declare there are no competing interests.

Nicole Miller conceived and designed the experiments, performed the experiments, analyzed the data, prepared figures and/or tables, authored or reviewed drafts of the paper, and approved the final draft.

Paul Maneval, Carrie Manfrino and Thomas K. Frazer conceived and designed the experiments, authored or reviewed drafts of the paper, and approved the final draft.

Julie L. Meyer conceived and designed the experiments, analyzed the data, prepared figures and/or tables, authored or reviewed drafts of the paper, and approved the final draft.

The following information was supplied regarding the deposition of DNA sequences:

Sequencing reads with primers and adapters removed are available at NCBI’s Sequence Read Archive, Bioproject PRJNA495377.

The following information was supplied regarding data availability:

All metadata and R scripts used for analysis are available at GitHub: https://github.com/meyermicrobiolab/Microbiomes-of-nursery-reared-Acropora-cervicornis.

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
