# Peer review of "Spatial distribution of microbial communities among colonies and genotypes in nursery-reared Acropora cervicornis"

_PeerJ, doi:10.7717/peerj.9635_

## Round 0.1 · original submission · Minor Revisions

Overall, your manuscript is well-written, clear, and generally well received. The two reviewers raised a number of concerns, though they are relatively minor and do not require further experiments or wet lab work.

Reviewer 1 ·

Basic reporting

The paper is well-written.

Some additional background is needed in the introduction:
it is confusing that you start off reference florida when the study takes place in the cayman islands.

I would also mention why cervicornis is especially easy to grow like this.

A discussion of Gignoux-Wolfsohn et al. 2017 should be added as they found a strong effect of presumed genotype (referred to as colony in the paper becuase they did not actually genotype) on A. cervicornis microbiomes. Although, many of those corals were dominated by Endozoicomonas. This difference and implications would be good to discuss as well.
S. Gignoux-Wolfsohn, F. M. Aronson, S. V. Vollmer, Complex interactions between potentially pathogenic, opportunistic, and resident bacteria emerge during infection on a reef-building coral. FEMS microbiology ecology 93, (2017).

Experimental design

The sampling strategy and methods are well-described.

A few comments on methods:
104- should be and yellow

106- were these wild colonies?

159: Did you do one analysis with all 3 of your factors? or separate?

Validity of the findings

Findings are well-described.

Comments on the results:
Missing figure legends?

Figure 4: Would be nice if this were all rickettsiales asvs. Given that there are only 11 I dont think it would significantly detract from your point and would allow the reader to make more clear comparisons between figure 3 and figure 4
Its also unclear if the y axis on this figure refers to total abundance or abundance within the rickettsiales?

177-184: its confusing that you are discussing most abundant asvs but referencing a figure where everything is grouped by order.

228-230: Would you expect rerunning the pipeline to produce different results? how much variation was there?

The discussion needs a little more work to place your findings in the context of the previous literature.

281: Also, they sampled differently, there could be differences you aren't capturing using syringe sampling. This sampling strategy should be discussed.

303: But Gignoux-Wolfsohn 2020 used V6 and found high abundance of Rickettsiales.

288-304: There is a lot of discussion of A. palmata that seems out of place given that this was not the objective of this study.

It is also interesting that you are sampling them right next to each other removing the environmental variation that confounds natural studies of genotypes. It would be interesting to have more of a discussion of the effect of site as found in Chu and Vollmer and Gignoux-Wolfsohn and Vollmer 2014

The discussion needs another paragraph focused on your finding of genotype-specificity of the dominant strains of Rickettsia as seen in Figure 4, espeically the finding that all samples across colonies of the green genotype were dominated by the same ASV. This is an interesting finding! To me it suggests a closer relationship between these rickettsia and the coral host. You can delve into more of a discussion of microbial symbionts of corals (both beneficial and pathogenic). What does this tell you about when these corals are aquiring the rickettsia?
see M. J. Neave et al., Differential specificity between closely related corals and abundant Endozoicomonas endosymbionts across global scales. The ISME journal, (2016).

·

Basic reporting

In this study, the Authors analyzed the microbial community composition of a keystone and endangered coral species, Acropora cervicornis, after sampling colonies that were reared in a Caribbean ocean nursery. Based on 16S rRNA gene metabarcoding, the Authors identified that bacterial communities did not display significant spatial heterogeneity across single colonies, but that they were influenced by the host genotype (with one bacterial genus dominated most samples). This work provides interesting information about the microbial communities associated with A. cervicornis and is a useful baseline for future studies.

Overall, the manuscript is clear, well written and structured, and the language is professional. Figures are relevant; raw data as well as scripts used for analysis are available.

The introduction flows well and is appropriately referenced, but could benefit from some additional information and minor revisions:

Lines 71-73: The allusion to coral microbiome manipulation comes a little bit abruptly and is not really the scope of the present study. While still useful to mention it, it would be interesting to expand on the roles of the coral microbiome and justify why it is important to characterize it (which would emphasize the value of this study).

Line 79: The Authors refer to coral “microbial communities”, which would actually encompass not only bacteria and archaea (assessed here), but also other microbes. As for example Symbiodiniaceae communities were not investigated, it would be more accurate to refer to “prokaryotic communities” or “bacterial communities” (when relevant) throughout the manuscript.

Lines 79-86: Either here or in the discussion, the Authors could emphasize why it is important to investigate spatial heterogeneity of bacterial communities in corals. If the communities are homogeneous, it would enable to collect fewer samples across the colony and hence reduce the stress of sampling. If the community composition varies across the colony, then it would be important to collect a sufficient number of samples from different locations on the same host, otherwise results wouldn’t be representative and within-colony differences could bias interpretations.

Line 86: It could be relevant to mention that some studies have reported a strong association between the coral host genotype and its bacterial community composition (for example: Glasl et al. 2019 PeerJ; or Hester et al. 2016 ISME - a review in which the authors listed studies having found an effect of coral species or environment on coral-associated bacterial communities).

Experimental design

Research questions are well-defined and most methods clearly explained, however the points outlined below require further explanations. Importantly, the Authors haven’t specified how they dealt with potential bacterial contaminants (as they didn’t seem to have done DNA extraction controls). If that is the case, this shortcoming should be acknowledged. Also, it is not indicated whether read counts were normalized prior to statistical analyses, which could bias the results given that samples had varying sequencing depths.

Line 115: What was the purpose of the agitation with the syringe? Was it to clear off the tissue from potential debris, trigger mucus release, or else…?

Line 120: Seawater was excluded from the samples by decanting, however it is unlikely that microorganisms present in seawater were entirely removed from samples. It would have been interesting to sample the seawater as well, and assess which part of the bacterial microbiome was coral-specific vs derived from the environment. This could perhaps be commented on?

Line 124: Was an equal amount of each sample used for DNA extraction (if so, how much)?

Lines 133-134: Negative PCR controls were performed, however there is no mention of DNA extraction controls. This is an important procedure to identify laboratory/reagent contaminants and is becoming a critical inclusion in microbiome studies (see Slater et al. 2014 BMC Biology; de Goffau et al. 2018 Nature Microbiology; Davis et al. 2018 Microbiome). If the Authors did not perform blank DNA extractions (and sequenced them alongside samples) to identify and remove contaminants, this should be explicitly acknowledged.

Lines 151 and further: Was any variance stabilization method (such as with DESeq2 or rarefaction) applied on the data to account for the different read counts across samples? Not normalizing the read counts could greatly impact and bias the results (especially since samples fell within a large range of counts, as outlined on line 168).

Line 160: Please specify that ANCOM stands for “Analysis of Composition of Microbiomes”

Validity of the findings

Results are nicely reported and discussed, and the conclusion clearly summarizes the findings of the study. Could the Authors please address the following points?

Lines 174-175: Was a pairwise comparison performed to check whether the difference in microbiome composition between the Y and the G/R was statistically significant?

Lines 209 and further: Did the Authors try to conduct the same analysis pipelines by removing these very abundant Rickettsiales from the samples? Sometimes rare members of the microbiome play important roles. The Authors specified on lines 232-234 that differences between coral colonies of different genotypes were based on microbes with low relative abundance. Such an analysis (with Rickettsiales removed from the dataset) would therefore be relevant to enable patterns to emerge that are based on low-abundance bacteria.

Lines 272-273: As the Authors outlined on the previous lines, it is interesting that the microbiome seems relatively uniform within A. cervicornis colonies and if so, stress of sampling could be reduced in future studies by collecting fewer samples while still obtaining a representative microbial community. However, the conclusion needs to be tempered, as patterns observed in one or two species cannot be directly extrapolated to all acroporid corals. This study in particular considered nursery-reared corals, which were hung on a structure and not growing on the reef as natural colonies would be. This in itself might lead to different microbial signatures and it would be relevant to repeat a similar study with wild A. cervicornis corals to confirm the present results. Moreover, other acroporid species (some of which have different morphology/physiology, or individuals sampled from another location) might possess microbial structures that are not consistent with those observed in this study. Hence, lack of spatial heterogeneity should not be assumed for most acroporid corals, at least without further evidence.

Lines 284-285: Maybe the Authors could suggest that more studies on spatial heterogeneity of microbial communities need to be undertaken across numerous coral species to verify these conclusions. If colonies are large enough, taking more than 3 samples per location on the colony would be a more robust design.

It could be useful to suggest that future studies should investigate the microbial community composition of corals transferred to the reef and compare how they relate to nursery-reared communities. Perhaps the success of individual corals could be linked with the composition of their microbiome, or some detrimental shifts could be identified in colonies that perform less well?

---

## Round 0.2 · accepted · Accept

I appreciate the time you have spent on revising your manuscript. It has resulted in an excellent study.

---

## Author Rebuttal · Round 0.2

**College of Agricultural and Life Sciences**
Soil and Water Sciences Department

Genetics Institute 304
PO Box 103610
Gainesville, FL 32611
352-273-8189

July 2 2020

Dear Dr. Rappe and reviewers:

On behalf of all the authors, I thank the editor and both reviewers for their comments and suggestions for our manuscript titled "Spatial distribution of microbial communities among colonies and genotypes in nursery-reared *Acropora cervicornis*". The subsequent revisions have greatly improved the manuscript. This includes the incorporation of several key references, yielding a more comprehensive introduction and discussion. We also clarified certain aspects of the methods and analysis as suggested by the reviewers. Please find attached a detailed point-by-point response to each reviewer's comments. Thank you very much for your consideration of this manuscript for publication in PeerJ.

Sincerely,

Julie L Meyer
Assistant Professor

**Editor comments (Michael Rappe)**

MINOR REVISIONS

Overall, your manuscript is well-written, clear, and generally well received. The two reviewers raised a number of concerns, though they are relatively minor and do not require further experiments or wet lab work.

***Note to reviewers: all line numbers are from the no-markup version. In the tracked changes document, the continuous line numbers are incorrect (this is a known issue with Microsoft Word that isn't fixable by the user).

Reviewer 1
**Basic reporting**
The paper is well-written.

Some additional background is needed in the introduction:
it is confusing that you start off reference florida when the study takes place in the cayman islands.

Agreed. The reference to the Florida Reef Tract was removed from the first sentence of the introduction (lines 44-45) In addition, lines 55-57 were modified to decrease the emphasis on Florida as follows: "For example, coral restoration projects have resulted in the outplanting of tens of thousands of nursery-reared *Acropora* corals on Florida reefs each year."

I would also mention why cervicornis is especially easy to grow like this.

Agreed. The following sentences were added to the second paragraph of the introduction (lines 63-68): "This improved growth and decreased mortality of *A. cervicornis* in ocean nurseries in comparison corals attached to reefs is due in part to reduced predation by the corallivorous snail *Coralliophila abbreviata* and the fireworm *Hermodice carunculate* [9]. In addition to the direct effects of predation on colony health, *C. abbreviata* is a known vector for white-band disease [13] and *H. carunculata* is a known vector for the coral pathogen *Vibrio shilohi* [14]."

A discussion of Gignoux-Wolfsohn et al. 2017 should be added as they found a strong effect of presumed genotype (referred to as colony in the paper becuase they did not actually genotype) on A. cervicornis microbiomes. Although, many of those corals were dominated by Endozoicomonas. This difference and implications would be good to discuss as well.
S. Gignoux-Wolfsohn, F. M. Aronson, S. V. Vollmer, Complex interactions between potentially pathogenic, opportunistic, and resident bacteria emerge during infection on a reef-building coral. FEMS microbiology ecology 93, (2017).

Good call. This aspect was missed because the main focus of the Gignoux-Wolfsohn paper is microbiome dynamics during disease transmission. We have added the following sentence to the last paragraph of the introduction (lines 102-104): "However, variation of bacterial communities in *A. cervicornis* from Panama used in a disease transmission study demonstrated that colony had a stronger effect than collection site [41]."

**Experimental design**
The sampling strategy and methods are well-described.

A few comments on methods:
104- should be and yellow

We replaced "or" with "and". (New line number 117)

106- were these wild colonies?

Yes. We have clarified this in the following sentence (lines 117-118): "Colonies of these genotypes were added to the CCMI nursery in 2012 as fragments from local populations."

159: Did you do one analysis with all 3 of your factors? or separate?

Differential abundance of bacterial families was determined for only coral genotype.

**Validity of the findings**
Findings are well-described.

Comments on the results:
Missing figure legends?

This comment is unclear. The figure legends appear on top of the figures in the pdf for review. Figure 3 is large and the legend appears on the page before it – is that what you mean?

Figure 4: Would be nice if this were all rickettsiales asvs. Given that there are only 11 I dont think it would significantly detract from your point and would allow the reader to make more clear comparisons between figure 3 and figure 4
Its also unclear if the y axis on this figure refers to total abundance or abundance within the rickettsiales?

The six Rickettsiales ASVs classified as genera other than MD3-55 were very low abundance (the highest average abundance on non-MD3-55 ASVs was 0.04% and it went down from there). This is indicated in Lines 234-235: "The remaining 9 Rickettsiales ASVs collectively had an average relative abundance well below 1%."

Figure 4 shows the relative abundance of the MD3-55 ASVs relative to the whole dataset. The legend for Figure 4 was clarified as follows: "Proportion of amplicon sequence variants classified as the Rickettsiales genus MD3-55 relative to all ASVs in the communities in colonies of green (G), red (R), and yellow (Y) coral genotypes of nursery-reared *Acropora cervicornis*."

177-184: its confusing that you are discussing most abundant asvs but referencing a figure where everything is grouped by order.

We have clarified this sentence as follows (lines 197-198): "The most abundant bacterial orders detected were Rickettsiales, Synechococcales, Vibrionales, and an unclassified order of Alphaproteobacteria (Figure 3)."

228-230: Would you expect rerunning the pipeline to produce different results? how much variation was there?

Rerunning the dada2 pipeline will create slightly different ASV tables that differ in the number of total ASVs detected – very low abundance ASVs will sometimes be included and sometimes be excluded as potential sequencing errors. However, the counts should not change.

Reviewer: Katarina Damjanovic
*Basic reporting*
In this study, the Authors analyzed the microbial community composition of a keystone and endangered coral species, Acropora cervicornis, after sampling colonies that were reared in a Caribbean ocean nursery. Based on 16S rRNA gene metabarcoding, the Authors identified that bacterial communities did not display significant spatial heterogeneity across single colonies, but that they were influenced by the host genotype (with one bacterial genus dominated most samples). This work provides interesting information about the microbial communities associated with A. cervicornis and is a useful baseline for future studies.

Overall, the manuscript is clear, well written and structured, and the language is professional. Figures are relevant; raw data as well as scripts used for analysis are available.

The introduction flows well and is appropriately referenced, but could benefit from some additional information and minor revisions:

Lines 71-73: The allusion to coral microbiome manipulation comes a little bit abruptly and is not really the scope of the present study. While still useful to mention it, it would be interesting to expand on the roles of the coral microbiome and justify why it is important to characterize it (which would emphasize the value of this study).

Thank you for this observation. We have the following additional sentences to the third paragraph of the introduction (lines 78-83):
"Several recent review papers [24, 26, 31-33] have detailed the potential roles that the microbiome plays in the overall health of the coral holobiont, including protection against pathogens, tight recycling of nutrients within the holobiont, and nitrogen fixation, which benefits the photosynthetic dinoflagellate symbionts, and there is growing interest in designer microbes and probiotic strains to mitigate loss of coral reefs [26, 27, 34, 35]."

Line 79: The Authors refer to coral "microbial communities", which would actually encompass not only bacteria and archaea (assessed here), but also other microbes. As for example Symbiodiniaceae communities were not investigated, it would be more accurate to refer to "prokaryotic communities" or "bacterial communities" (when relevant) throughout the manuscript.

True, "microbial community" could encompass not only Bacteria and Archaea, but also Symbiodiniaceae, fungi, ciliates, viruses, etc. However, "microbial" is not inherently an all-inclusive term (like holobiont) and is very commonly used to mean just bacterial and archaeal communities throughout the literature. Replacing "microbial" with "bacterial" would not be accurate in most cases, so it is easier to say "microbial" for "bacterial and archaea". For clarity, we have amended the abstract methods as follows: "We characterized the bacterial and archaeal community composition of *A. cervicornis* corals in a Caribbean nursery to determine the heterogeneity of the microbiome within and among colonies." That way, it will be clear from the beginning what fractions of the microbial community were targeted.

Lines 79-86: Either here or in the discussion, the Authors could emphasize why it is important to investigate spatial heterogeneity of bacterial communities in corals. If the communities are homogeneous, it would enable to collect fewer samples across the colony and hence reduce the stress of sampling. If the community composition varies across the colony, then it would be important to collect a sufficient number of samples from different locations on the same host, otherwise results wouldn't be representative and within-colony differences could bias interpretations.

This idea was addressed in the first paragraph of the discussion, but we have expanded this in the introduction and discussion to further emphasize the importance of this observation.

Introduction (lines 89-91):
"First, we assess whether a single sample is representative of the microbiota across an entire colony, information that is useful for both researchers and resource managers that strive to reduce stressors to already threatened species."

Discussion (lines 289-295):

"This has important implications for researchers and resource managers who are concerned with how many samples are appropriate to take per colony and how to minimize sampling to reduce stress to colonies. It is also important to note that even with the heavy sampling that was performed here, namely nine samples taken in one day from a basketball-sized colony, no visible stress to the colonies was discernable. These colonies were observed during regular maintenance of the *in situ* nursery and roughly a year after sampling, all of the colonies were thriving."

Line 86: It could be relevant to mention that some studies have reported a strong association between the coral host genotype and its bacterial community composition (for example: Glasl et al. 2019 PeerJ; or Hester et al. 2016 ISME - a review in which the authors listed studies having found an effect of coral species or environment on coral-associated bacterial communities).

Discussion of the Glasl and Hester papers were added to the discussion (lines 367-372). "This is consistent with recent work demonstrating that while the genus *Endozoicomonas* was predominant among *Acropora tenuis* bacterial communities, individual ASVs of *Endozoicomonas* were highly host genotype-specific [76]. Similarly, *Endozoicomonas* strains were identified as sporadic symbionts in the Pacific Line Islands [78], where *Endozoicomonas* was found on *Porites lutea* from only one island out of six islands sampled."

*Experimental design*
Research questions are well-defined and most methods clearly explained, however the points outlined below require further explanations. Importantly, the Authors haven't specified how they dealt with potential bacterial contaminants (as they didn't seem to have done DNA extraction controls). If that is the case, this shortcoming should be acknowledged. Also, it is not indicated whether read counts were normalized prior to statistical analyses, which could bias the results given that samples had varying sequencing depths.

Line 115: What was the purpose of the agitation with the syringe? Was it to clear off the tissue from potential debris, trigger mucus release, or else…?

This is just to trigger mucus release.

Line 120: Seawater was excluded from the samples by decanting, however it is unlikely that microorganisms present in seawater were entirely removed from samples. It would have been interesting to sample the seawater as well, and assess which part of the bacterial microbiome was coral-specific vs derived from the environment. This could perhaps be commented on?

True, the samples likely contain at least some seawater microbes. We did not process seawater samples along with the coral samples in this study, but it is well established that the surface mucus layer of corals is very distinct from seawater. The Rickettsiales bacteria that were so predominant here are likely intracellular and therefore not likely from seawater.

Line 124: Was an equal amount of each sample used for DNA extraction (if so, how much)?

This detail has been added to the methods (lines 138-140): "DNA was extracted from up to 0.5 ml of mucus and tissue using a DNeasy Powersoil Kit (Qiagen, Germantown, MD) according to the manufacturer's instructions, including bead beating for 10 minutes."

Lines 133-134: Negative PCR controls were performed, however there is no mention of DNA extraction controls. This is an important procedure to identify laboratory/reagent contaminants and is becoming a critical inclusion in microbiome studies (see Slater et al. 2014 BMC Biology; de Goffau et al. 2018 Nature Microbiology; Davis et al. 2018 Microbiome). If the Authors did not perform blank DNA extractions (and sequenced them alongside samples) to identify and remove contaminants, this should be explicitly acknowledged.

Thankfully, this is one of the last studies from our lab that does not have extraction controls. We have added these sentences to the methods (lines 140-144):
"As extraction controls were not collected when samples were processed in early 2018, we acknowledge the potential for contamination from lab reagents [43], either in the extraction kit or in subsequent PCR and cleanup reagents. The processing of extraction controls has since become standard practice in our lab beginning in mid-2018."

Lines 151 and further: Was any variance stabilization method (such as with DESeq2 or rarefaction) applied on the data to account for the different read counts across samples? Not normalizing the read counts could greatly impact and bias the results (especially since samples fell within a large range of counts, as outlined on line 168).

Line 174
We performed a centered log-ratio transformation, as recommended for compositional data by Gloor et al. 2017 (https://www.frontiersin.org/articles/10.3389/fmicb.2017.02224/full#h3)

Line 160: Please specify that ANCOM stands for "Analysis of Composition of Microbiomes"

Good catch. Lines 178-181: "Differential abundance of microbial families among coral genotypes was determined with Analysis of Composition of Microbiomes (ANCOM) [56]…"

*Validity of the findings*
Results are nicely reported and discussed, and the conclusion clearly summarizes the findings of the study. Could the Authors please address the following points?

Lines 174-175: Was a pairwise comparison performed to check whether the difference in microbiome composition between the Y and the G/R was statistically significant?

It is not clear how this would be achieved. The ANOSIM tested whether communities were statistically different among host genotypes.

Lines 209 and further: Did the Authors try to conduct the same analysis pipelines by removing these very abundant Rickettsiales from the samples? Sometimes rare members of the microbiome play important roles. The Authors specified on lines 232-234 that differences between coral colonies of different genotypes were based on microbes with low relative abundance. Such an analysis (with Rickettsiales removed from the dataset) would therefore be relevant to enable patterns to emerge that are based on low-abundance bacteria.

The results of the differential abundance analysis did not pick out Rickettsiales as important to the differences between coral genotypes. Therefore, that analysis did reveal the relatively rare members that contributed to the differences between host genotypes. We also plotted the stacked bars without Rickettsiales, but this wasn't any more informative than the full dataset. The main non-Rickettsiales differences are visible in Figure 3 and are discussed in the text.

Lines 272-273: As the Authors outlined on the previous lines, it is interesting that the microbiome seems relatively uniform within A. cervicornis colonies and if so, stress of sampling could be reduced in future studies by collecting fewer samples while still obtaining a representative microbial community. However, the conclusion needs to be tempered, as patterns observed in one or two species cannot be directly extrapolated to all acroporid corals. This study in particular considered nursery-reared corals, which were hung on a structure and not growing on the reef as natural colonies would be. This in itself might lead to different microbial signatures and it would be relevant to repeat a similar study with wild A. cervicornis corals to confirm the present results. Moreover, other acroporid species (some of which have different morphology/physiology, or individuals sampled from another location) might possess microbial structures that are not consistent with those observed in this study. Hence, lack of spatial heterogeneity should not be assumed for most acroporid corals, at least without further evidence.

Agreed. We have modified the end of the first paragraph of the Discussion as follows (lines 299-304): "Thus, for Caribbean acroporid corals, it appears that microbial community composition is relatively uniform at the colony scale. Since both of these coral species are critically endangered, this means that sampling for microbiota can be minimized to reduce impact to the coral host. However, this remains to be tested more broadly across different coral species, and especially in wild populations when available, rather than in the relatively sheltered nursery-reared colonies sampled here."

Lines 284-285: Maybe the Authors could suggest that more studies on spatial heterogeneity of microbial communities need to be undertaken across numerous coral species to verify these conclusions. If colonies are large enough, taking more than 3 samples per location on the

colony would be a more robust design.

Agreed. The first part of this comment – testing in more coral species - was addressed in the previous comment. Suggesting specific sampling schemes feels like overstepping.

It could be useful to suggest that future studies should investigate the microbial community composition of corals transferred to the reef and compare how they relate to nursery-reared communities. Perhaps the success of individual corals could be linked with the composition of their microbiome, or some detrimental shifts could be identified in colonies that perform less well?

Indeed! This has been the focus of several funding proposals by our group and we have a project in progress that will address this.